# Multiple reaction pathway on alkaline earth imide supported catalysts for efficient ammonia synthesis

Zichuang Li[1], Yangfan Lu [2], Jiang Li [3], Miao Xu[4], Yanpeng Qi [5,6,7], Sang-Won Park [3], Masaaki Kitano [3] ✉, Hideo Hosono [3] ✉, Jie-Sheng Chen [1] & Tian-Nan Ye [1] ✉

The tunability of reaction pathways is required for exploring efficient and low cost catalysts for ammonia synthesis. There is an obstacle by the limitations arising from scaling relation for this purpose. Here, we demonstrate that the alkali earth imides (*Ae*NH) combined with transition metal (TM = Fe, Co and Ni) catalysts can overcome this difficulty by utilizing functionalities arising from concerted role of active defects on the support surface and loaded transition metals. These catalysts enable ammonia production through multiple reaction pathways. The reaction rate of Co/SrNH is as high as 1686.7 mmol·$g_{Co}^{-1}$·$h^{-1}$ and the TOFs reaches above 500 $h^{-1}$ at 400 °C and 0.9 MPa, outperforming other reported Co-based catalysts as well as the benchmark Cs-Ru/MgO catalyst and industrial wüstite-based Fe catalyst under the same reaction conditions. Experimental and theoretical results show that the synergistic effect of nitrogen affinity of 3d TMs and in-situ formed $NH^{2-}$ vacancy of alkali earth imides regulate the reaction pathways of the ammonia production, resulting in distinct catalytic performance different from 3d TMs. It was thus demonstrated that the appropriate combination of metal and support is essential for controlling the reaction pathway and realizing highly active and low cost catalysts for ammonia synthesis.

Ammonia ($NH_3$) has been one of the most critical intermediates for various chemicals and fertilizers[1,2]. Recently, $NH_3$ has also attracted attention as a renewable energy carrier because of its high energy density (22.5 kJ·$g^{-1}$) and hydrogen content (17.6 wt%)[3,4]. The industrial $NH_3$ is majorly produced by Haber–Bosch process that requires high temperature (400–600 °C) and pressure (20–40 MPa) conditions[5]. In ammonia synthesis, the dissociation of nitrogen molecules is regarded as the most challenging step due to its stable N≡N triple bond[6–8]. Therefore, many of the previous studies have focused on weakening the N≡N bond during catalytic reactions. For example, in Fe and Ru-based catalysts, the addition of basic promoters, such as alkali and alkaline earth metal oxides, have been intensively investigated because they give rise to electron transfer from the catalysts to the antibonding (π*) state of nitrogen molecules, weakening the N≡N bond

[1]Frontiers Science Center for Transformative Molecules, School of Chemistry and Chemical Engineering, Shanghai Jiao Tong University, Shanghai 200240, China. [2]College of Materials Science and Engineering, National Engineering Research Center for Magnesium Alloys, Chongqing University, Chongqing 400044, China. [3]Materials Research Center for Element Strategy, Tokyo Institute of Technology, 4259 Nagatsuta, Midori-ku, Yokohama 226-8503, Japan. [4]State Key Laboratory of Space Power Sources, Shanghai Institute of Space Power-Sources, Shanghai 200245, China. [5]School of Physical Science and Technology Shanghai Tech University, Shanghai 201210, China. [6]ShanghaiTech Laboratory for Topological Physics, ShanghaiTech University, Shanghai 201210, China. [7]Shanghai Key Laboratory of High-resolution Electron Microscopy, ShanghaiTech University, Shanghai 201210, China. ✉e-mail: kitano.m.aa@m.titech.ac.jp; hosono@mces.titech.ac.jp; ytn2011@sjtu.edu.cn

through electron donation mechanism[9–18]. However, the promotion effect of basic supports is not sufficient to facilitate ammonia synthesis under mild conditions due to the limited electron donation ability. Another approach to lower the apparent activation energy is to use electride as support materials[19–29], such as the $12CaO \cdot 7Al_2O_3$ (C12A7:e$^-$) electride. In contrast to the traditional promoters, C12A7:e$^-$ is characterized by the co-existence of high electron density with low work function, chemical and thermal stability, and reversible exchangeability between anionic electron and hydrogen, realizing much stronger electron transfer from support to active transition metals and robustness to hydrogen poisoning. Consequently, the rate-determine step of ammonia synthesis was shifted from $N_2$ dissociation to the $NH_x$ formation in Ru/C12A7:e$^-$, contributing to suppress the reaction activation energy[30].

In either traditional or electride-based catalysts, their transition metal sites are responsible for catalytic reactions, and the catalytic activity strongly depends on the magnitude of the interaction between nitrogen and transition metals, which is widely known as the scaling relation[31]. In this reaction mechanism, Ru is located at around the optimal point and thus show excellent catalytic activity, while low cost 3d transition metals, such as Fe, Co and Ni, were less effective under mild reaction condition[32–38]. Recently, the combination of hydride and transition metals were focused on overcoming the bottlenecks. Chen et al.[39,40] showed that the scaling relation, as well as the reaction path of ammonia synthesis, were shifted by using the transition metals loaded LiH (TM/LiH), in which $N_2$ molecule initially dissociated to N* on the surface of TMs and then transferred to LiH support, forming the Li-$NH_x$ species as intermediates. Subsequently, the $NH_x$ species undergo further hydrogenation to form $NH_3$. Such reaction path has circumvented the limitation of the scaling relation by functionalizing the lattice hydrogen of LiH supports. Similar strategy was also proved to be effective on $Fe/TiO_{2-x}H_y$ catalyst[41], in which $N_2$ and $H_2$ are both dissociated on Fe and reacted with each other to form $NH_3$ in the oxygen vacancy site through a spillover process. Recently, transition metal nitrides, especially $Co_3Mo_3N$, have also received great attention for ammonia synthesis since the nitrogen defects are reported to be effective for the nitrogen molecule activation through a Mars-van Krevelen mechanism, which can significantly weaken the N≡N bond and lower its dissociation barrier[42–46]. For each case, the modified reaction path from TMs to support material leads to a low activation energy barrier, allowing $NH_3$ synthesis at low temperatures.

Based on a concept of the synergistic effect between TMs and supports' defects, we developed TM/ReN (TMs = Co and Ni; Re = La and Ce) catalysts for ammonia synthesis[47–49]. Among them, the dual active site strategy was utilized, enabling the $N_2$ activation at the nitrogen vacancy ($V_N$) sites of ReN support. That is, $V_N$ has trapped electrons with low work function reflecting the property of rare earth. This is the reason why $V_N$ in ReN works well as sites for $N_2$ activation. Ni/ReN successfully overcame the scaling limitations associated with the weak nitrogen binding energy of Ni. In Co/CeN, the in situ formed $V_N$ sites provide sites for $N_2$-activation as well. $V_N$ in ReN is an anionic defect and featured by electron-rich and low-work function properties[47]. These results inspire us to extend this idea to other system; the electron-rich vacancies would be effective in improving the catalytic activity of 3d TMs themselves that facilitate dissociative $N_2$ adsorption aside from defects driven pathway, giving rise to the associative-dissociative concerted mechanism. So far, most of the defect-driven ammonia synthesis catalysts utilize anionic defects of single atoms, such as $O^{2-}$, $H^-$ and $N^{3-}$[41,50,51], while the functionalities of polyanions, such as $NH^{2-}$, remained unexplored despite of their intriguing properties associated with its large anionic volume that would be favorable for $N_2$ activation with expectation for allowing various adsorption geometry for adsorbed $N_2$ and enhanced trapping efficiency of lower work function electrons. We are therefore motivated to explore the materials having larger defect sites, enabling to control the reaction

pathway of ammonia synthesis and further improve its catalytic activity.

Here, we report that alkaline earth imides, $Ae$NH ($Ae$ = Ca, Sr, Ba), is one of the ideal platforms to control the reaction pathway by utilizing its anionic defects. In contrast to the previously studied materials, the anionic defects of $Ae$NH are composed of polyanionic group ($NH^{2-}$), which offer much larger space to host nitrogen, i.e., the $NH^{2-}$ defect is 1.2 times as large as $N^{3-}$ defect in LaN. DFT calculation shows that defective SrNH has much lower work function property than previously studied CeN and LaN. As the results, the nitrogen adsorption properties of TMs are strongly affected by the in situ formed $NH^{2-}$ vacancy of $Ae$NH, enabling ammonia production through multiple reaction pathways. By loading 3d TMs (TM = Fe, Co and Ni), TM/$Ae$NH continuously produced ammonia and reached 779.2 mmol·$g_{Co}^{-1}$·h$^{-1}$ at 400 °C and 0.9 MPa for Co/SrNH, which is six times greater than previously studied Ni/LaN. Isotope experiments in combination with DFT calculations clarifies that the cooperation of the surface low work function (-2.0 eV) feature and the in situ formation of large sized $NH^{2-}$ vacancies on SrNH support gives rise to a dual pathway for ammonia synthesis over Co/SrNH catalyst. These discoveries show the introduction of large sized electron-rich anionic vacancy enables $N_2$ activation with low activation energy barrier, providing a material design strategy to realize highly efficient ammonia synthesis under mild reaction conditions.

## Results

Density functional theory (DFT) calculations were first performed to investigate the $NH^{2-}$ vacancy formation of the $Ae$NH ($Ae$ = Ca, Sr, Ba) (Fig. S1a). The calculated vacancy formation energy ($E_V$) for CaNH, SrNH, and BaNH are estimated to be BaNH (0.85 eV) < SrNH (1.92 eV) < CaNH (2.35 eV), indicating the formation of $NH^{2-}$ vacancy is more energy favored for BaNH and SrNH comparing to CaNH. Notably, $E_V$ of SrNH and BaNH are comparable to those of LaN (1.90 eV) and CeN (1.39 eV). The Bader charge of the $V_{NH}$ sites in CaNH, SrNH, and BaNH were calculated to be −1.24, −1.54 and −1.61 (Fig. S1b–d), respectively, which suggested that substantial electrons originated from the $NH^{2-}$ ion are accumulated in $V_{NH}$ sites of $Ae$NH[52], indicating that the defects also exhibit strong electron donation ability. Interestingly, $Ae$NH with surface $V_{NH}$ sites was found to show extremely low work function characteristics, which will be discussed below. We thus expect the easily formed electron-rich $NH^{2-}$ vacancy site can provide additional active sites for $N_2$ activation and promote a high catalytic performance in ammonia synthesis.

Subsequently, the catalytic performance of $Ae$NH for ammonia synthesis was investigated by loading Ni (Ni/$Ae$NH). The initial ammonia production rates of Ni/BaNH and Ni/SrNH are comparable (Fig. S2a), but much higher than that of Ni/CaNH catalyst. Meanwhile, BaNH support was not stable during the reaction and decomposed to unknown phases as indicated in XRD patterns (Fig. S2b–d), resulting an obvious degradation of catalytic activity (Fig. S2e). Thus, we focus SrNH as a model support in this work (Fig. S3). Figure 1a shows the catalytic activity of TM/SrNH as the function of temperature under 0.1 MPa. Among them, Co/SrNH shows the highest catalytic activity with the ammonia production rate of 5.0 mmol·$g_{cat.}^{-1}$·h$^{-1}$ at 340 °C (Table S1). The effluent $NH_3$ concentration for Co/SrNH reached thermodynamic equilibrium above 360 °C and 0.1 MPa pressure. The calculated activation energies ($E_a$) of the Co/SrNH and Fe/SrNH catalysts are -50 kJ·mol$^{-1}$ in the temperature range of 280–360 °C at 0.1 MPa, fairly smaller than Ni catalyst having $E_a$ = 90 kJ·mol$^{-1}$ (Fig. 1b). The higher $E_a$ of Ni/SrNH is attributed to the different catalytic mechanism from Co/SrNH, Fe/SrNH and Ni/SrNH. The reaction orders with respect to $N_2$, $H_2$, and $NH_3$ over TM/$Ae$NH catalysts are shown in Table 1 and Fig. S4. $N_2$ reaction orders (α) for all tested catalysts are in the range of 0.8 ~ 1.2, whereas the $H_2$ reaction orders (β) strongly depend on employed active metals. Compared to a bench mark

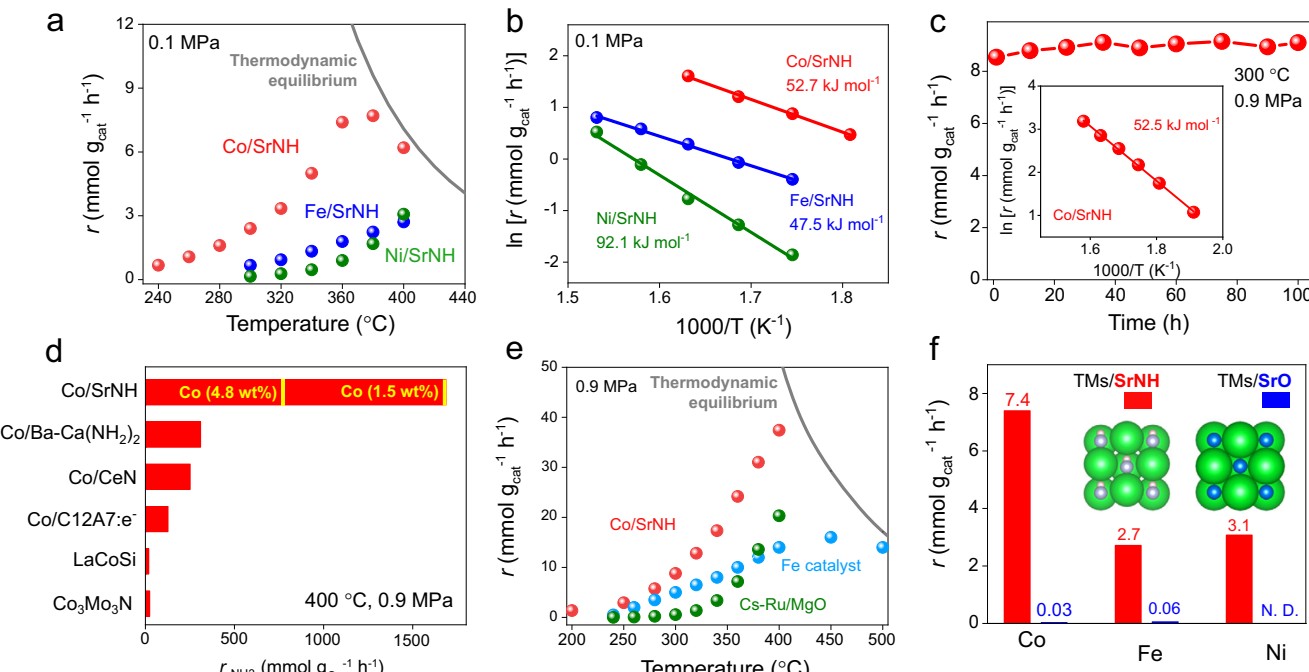

**Fig. 1 | Catalytic activity of *TM/Ae*NH catalysts. a** Temperature dependence of the NH$_3$ synthesis rates over Co/SrNH, Fe/SrNH and Ni/SrNH catalysts under 0.1 MPa. **b** Arrhenius plots for NH$_3$ synthesis over Co/SrNH, Fe/SrNH and Ni/SrNH catalysts at 0.1 MPa. **c** Stability test for NH$_3$ synthesis over Co/SrNH at 300 °C, 0.9 MPa. Insets: Arrhenius plots over Co/SrNH at 0.9 MPa. **d** Catalytic activity of different Co catalysts in NH$_3$ synthesis under 400 °C and 0.9 MPa. **e** Temperature dependence of the NH$_3$ synthesis rates over Co/SrNH, Cs-Ru/MgO, wüstite-based Fe catalysts under 0.9 MPa. **f** Catalytic activity of TMs-supported SrNH (red) and SrO (blue) catalysts for NH$_3$ synthesis at 0.1 MPa with Co- at 360 °C, Fe- and Ni- at 400 °C.

catalyst Cs−Ru/MgO with negative H$_2$ reaction orders, the positive value over TM/*Ae*NH catalysts indicate catalysts' robustness against hydrogen poisoning. It should be noted that the H$_2$ reaction order ($\beta$) of TM/*Ae*NH catalysts changed from 1.7 (Co/SrNH) and 1.6 (Fe/SrNH) to 0.2 (Ni/CaNH) and 0.1 (Ni/SrNH). Such a low H$_2$ reaction order of Ni catalysts can be attributed to slow consumption of dissociated H*, resulting in a high coverage of adsorbed H* on Ni surface. This is consistent with the relative larger E$_a$ (ca. 90 kJ·mol$^{-1}$) of Ni/CaNH (Fig. S5) and Ni/SrNH (Fig. 1b) compare to Co/SrNH and Fe/SrNH, which will be discussed later. Meanwhile, the large positive H$_2$ reaction orders ($\beta$) of Co/SrNH and Fe/SrNH would be expected to lead a favorable pressure effect for ammonia synthesis, which can be confirmed by the linearly enhanced NH$_3$ production rates over both Co/SrNH and Fe/SrNH catalysts under 0.9 MPa (Fig. S6). The Co/SrNH continuously produced ammonia at least for 100 h without clear degradation (Fig. 1c). High-angle annular dark-field scanning transmission electron microscopy (HAADF-STEM) images and corresponding Energy-dispersive X-ray spectroscopy (EDX) mapping results clearly demonstrated that Sr and N are uniformly dispersed on SrNH and the size of Co nanoparticle remained largely unchanged after long-term reaction (Figs. S7, S8), which indicates the stability of Co/SrNH. X-ray photoelectron spectroscopy (XPS) and Auger electron spectroscopy (AES) were further performed to check the surface element

change of Co/SrNH before and after reactions. The results show that the valence state of Co and Sr species remains unchanged while the surface N content slightly decreased after the reaction (Fig. S9), which suggest that the lattice NH$^{2-}$ may participate in the formation of NH$_3$. The E$_a$ of Co/SrNH under 0.9 MPa is largely unchanged with 0.1 MPa and much lower than that of conventional Ru based catalysts (133.0 kJ·mol$^{-1}$, Fig. S10), suggesting that the catalytic mechanism is retained (Fig. 1c inset).

The ammonia production rate of Co/SrNH reaches 1686.7 mmol·g$_{Co}$$^{-1}$·h$^{-1}$ at 400 °C and 0.9 MPa, which is much higher than those of reported Co- and Ni-based catalysts measured under a relative low space velocity (Fig. 1d, Table S2). It is noted that Co/SrNH with a low amount of Co loading (1.5 wt %) shows slightly lower reaction rates (Fig. S11a), but comparable TOFs to that of Co (4.8 wt %)/SrNH (Table S3), which should be ascribed to the smaller Co particle size as shown in Fig. S12. We are also aware that the apparent activation energy of Co/SrNH with a low amount of Co remains largely unchanged compared to Co (4.8 wt %)/SrNH (Fig. S11b), which indicated a similar reaction mechanism. The calculated turnover frequencies (TOFs) of Co (4.8 wt%)/SrNH is as high as 500 h$^{-1}$, also far exceed those for previous reported Co based catalysts[13–15,22,32,35,39,40,47,51,53–56] (Fig. S13, Table S3). Impressively, the activity of Co/SrNH even outperform the majority of reported Ru metal catalysts under similar reaction conditions[25,32,51,53,57–60] (Fig. S13, Tables S4, S5). In terms of NH$_3$ production rate, Co/SrNH outperforms the benchmark Cs-Ru/MgO catalyst and industrial wüstite-based Fe catalyst tested under the same reaction conditions (Fig. 1e). The specific activity of Co/SrNH (2.35 mmol·m$^{-2}$·h$^{-1}$, 400 °C, 0.9 MPa) is 5 times higher than those of our previously reported rare-earth metal nitride catalysts (Fig. S14). To unveil the high activity origin of Co/SrNH catalyst, we replaced SrNH support by SrO and test the ammonia synthesis under the same conditions. SrO was employed as the counter compound because they can be categorized to the related materials. If we consider NH$^{2-}$ as an anion unit, both SrO and SrNH can be regarded as the rock salt type

## Table 1 | Reaction orders of ammonia synthesis for various catalysts

| Catalyst | N$_2$ order ($\alpha$) | H$_2$ order ($\beta$) | NH$_3$ order ($\gamma$) |
|---|---|---|---|
| Fe/SrNH | 1.1 | 1.6 | −1.6 |
| Co/SrNH | 1.2 | 1.7 | −1.6 |
| Ni/SrNH | 1.2 | 0.1 | −1.2 |
| Ni/CaNH | 1.2 | 0.2 | −1.1 |
| Cs-Ru/MgO | 1.1 | −0.5 | −0.35 |

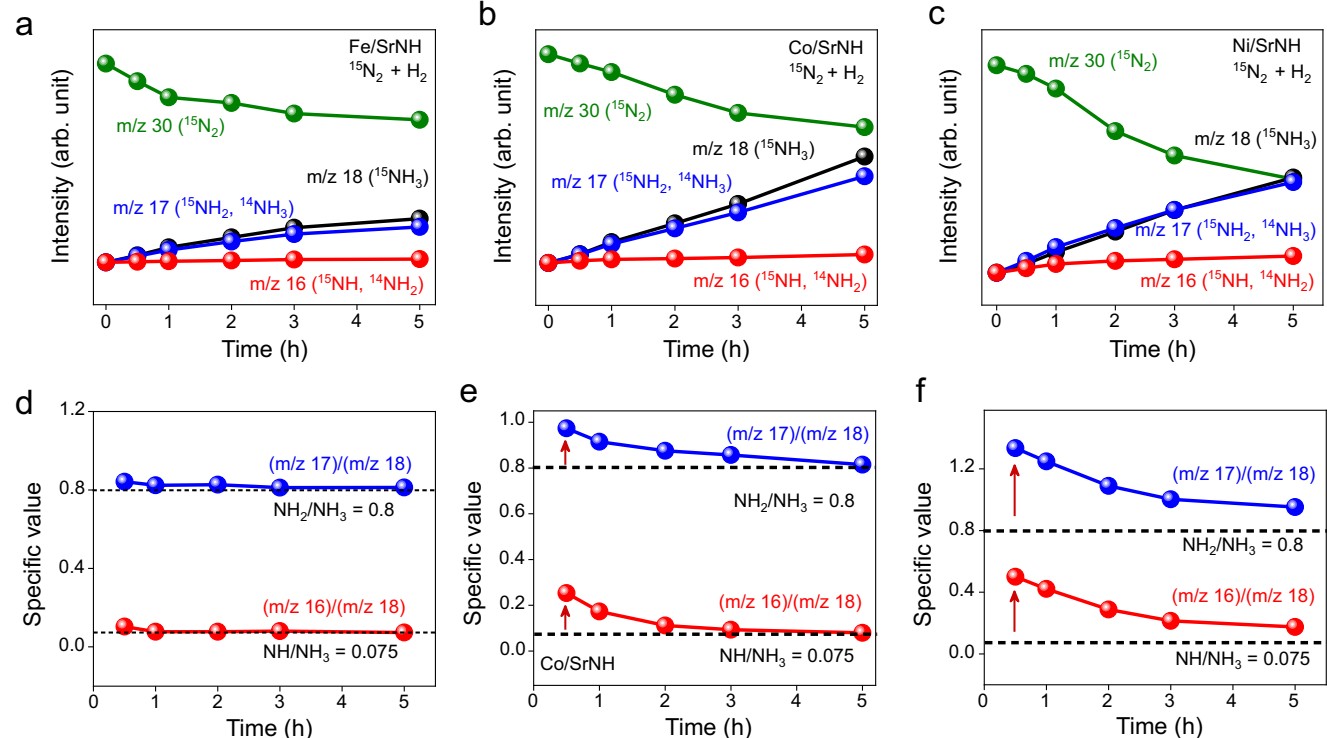

**Fig. 2 | $^{15}N_2/H_2$ isotopic experiments of TMs-SrNH catalysts.** Reaction time profiles (**a**–**c**) for NH$_3$ synthesis from $^{15}N_2$ and H$_2$, and (**d**–**f**) the ratio changes of $m/z$ 17/18 and 16/18 over fresh Fe/SrNH, Co/SrNH, and Ni/SrNH catalysts.

structure. Meanwhile, the two systems are distinct in terms of anion defect formation energy. In contrast to the relatively low NH$^{2-}$ defect formation energy of SrNH (1.92 eV), the defect formation energy of O$^{2-}$ of SrO reached 5.38 eV (Table S6), giving an ideal platform to investigate the effect of NH$^{2-}$ defect to its catalytic mechanism. The NH$_3$ production significantly decreased to near the detection limit over SrO supported Co and Fe catalysts, and even could not be detected for Ni/SrO, demonstrating that the NH$^{2-}$ defect plays critical role during ammonia production in TM/SrNH (Fig. 1f).

To unveil the importance of the lattice NH$^{2-}$ defects in the catalytic mechanism, the isotopic experiments were performed over TM/SrNH catalysts using $^{15}N_2/H_2$. With the increase of temperature, TM/SrNH continuously consumes $^{15}N_2$ and lattice $^{14}NH^{2-}$ of SrNH, producing $^{15}NH_3$ and $^{14}NH_3$. Therefore, the mass signals measured for $m/z$ = 18 ($^{15}NH_3$), 17 ($^{15}NH_2$, $^{14}NH_3$), and 16 ($^{15}NH$, $^{14}NH_2$) are gradually enhanced (Fig. 2a–c). The intensity ratios of $m/z$ = 17/18 and 16/18 over Co/SrNH and Ni/SrNH catalysts were larger than the theoretical values (assuming all NH$_3$ are derived from $^{15}N_2$) of 0.8 ($m/z$ 17/18) and 0.075 ($m/z$ 16/18) at the initial state (at 0.5 h), suggesting that the formation of NH$_3$ was derived from both $^{15}NH_3$ and $^{14}NH_3$ (Fig. 2b, c). Meanwhile, the $m/z$ = 17/18 and 16/18 intensities for the Fe/SrNH catalyst are very close to the theoretical value at the initial state because the formation of $^{14}NH_3$ is negligible (Fig. 2a). These results demonstrate that while the lattice $^{14}NH^{2-}$ of Co/SrNH and Ni/SrNH are involved in the catalytic cycle, its participation is smaller over Fe/SrNH catalyst. Owing to much stronger Fe-N interaction, $^{15}N_2$ and H$_2$ are both activated on the Fe surface. These results show the multiple catalytic mechanism for TM/SrNH depending on TM.

To furthermore investigate the catalytic mechanism, we subsequently conducted the isotope experiments employing N$_2$ and D$_2$. Ammonia (NH$_3$) and its isotopic species (ND$_3$, ND$_2$H, NDH$_2$, ND$_2$, NDH, ND, and NH$_2$) were detected with varying temperatures from 25 °C to 400 °C. It is noted that not only gas phase D$_2$, but also lattice H species participate to ammonia production. In Fig. 3a, b, the ammonia isotopic fragments were detected simultaneously over Fe/

SrNH and Co/SrNH above ca. 200 °C. It is in stark contrast to Ni/SrNH, in which the formation of ND$_3$ was delayed compared to other fragments, i.e., ammonia isotopic fragments were detected above ca. 380 °C (Fig. 3c). Considering the weak interaction between Ni and N, Ni is unlikely responsible for N$_2$ activation. Thus, such delayed $m/z$ = 20 (ND$_3$) signals may be attributed to the reaction of D* created on Ni with the lattice NH$^{2-}$ and subsequent formation of NH$^{2-}$ vacancy at SrNH support surface. The in situ generated NH$^{2-}$ vacancy sites can serve as the activation centers for N$_2$ and subsequent hydrogenation to NH$_3$. In contrast, Co and Fe show much higher nitrogen affinity, giving rise to simultaneous activation of N$_2$ and D$_2$ on their surface and immediate formation of ND$_3$. Accordingly, in $^{14}N/^{15}N$ isotopic exchange experiments ($^{14}N_2 + ^{15}N_2 \rightarrow ^{14}N^{15}N$), the reaction rate of N$_2$ isotope exchange over Co/SrNH (2.03 mmol·g$^{-1}$·h$^{-1}$) becomes 5 times higher than that for Ni/SrNH (0.35 mmol·g$^{-1}$·h$^{-1}$), and comparable to Fe/SrNH (2.36 mmol·g$^{-1}$·h$^{-1}$) (Fig. 3d). The apparent E$_a$ for Ni/SrNH was obtained as 156.5 kJ mol$^{-1}$, showing the highest value among the three catalysts (Fig. S15). These results suggest that the N$_2$ dissociation was preferred on Fe and Co metal, but unlikely on Ni metal, which is consistent with our hypothesis discussed above.

H$_2$-temperature programmed reaction (H$_2$-TPR) measurements were further conducted to elucidate the reaction pathway of ammonia synthesis over TM/SrNH catalysts. Note that the dissociated N* and/or formed NH$_x$ species remain on the surface of each used TM/SrNH catalyst. In used Fe/SrNH, the ammonia fragments (NH$_3$, NH$_2$ and NH) appeared at ca. 200 °C, consistent with the N$_2$/D$_2$ isotope experiments, suggesting that the Fe metal is mainly responsible for the N$_2$ and H$_2$ activation and subsequent NH$_3$ formation (Fig. 4a). In it, SrNH support plays smaller contributions, consistent with the $^{15}N/H_2$ isotopic results (Fig. 2a). Meanwhile, the desorption peak at ca. 200 °C was absent in Ni/SrNH due to the relatively weak nitrogen interaction of Ni metal (Fig. 4c), i.e., N$_2$ is unlikely activated on Ni metal. Instead, a desorption peak at a higher temperature region (ca. 350 °C) can be identified for Ni/SrNH, which is attributed to the reaction between the dissociated H* and the lattice NH$^{2-}$ in SrNH support (Fig. 4c). It should be emphasized

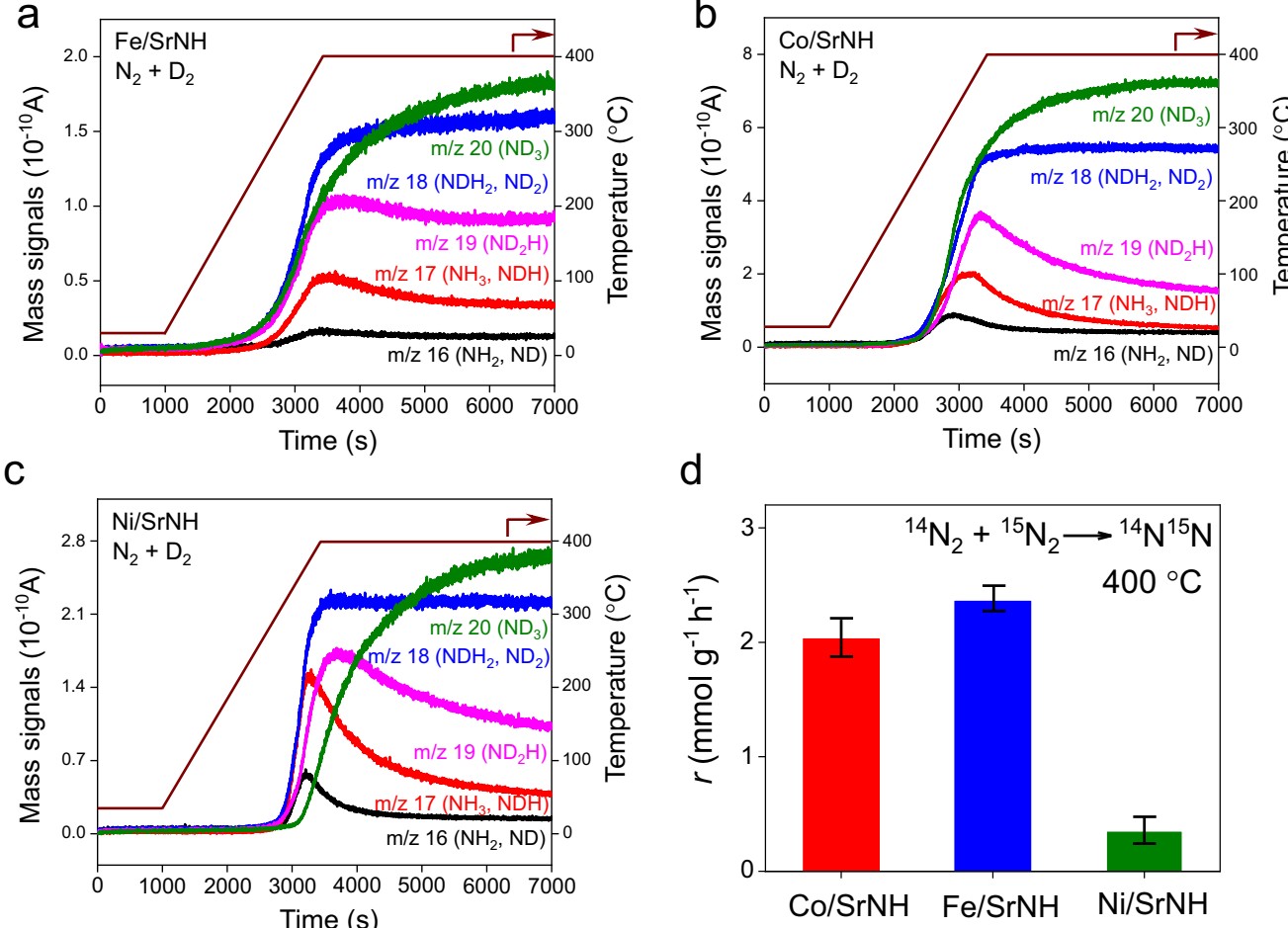

**Fig. 3 | N₂/D₂ and ¹⁴N/¹⁵N isotopic experiments of TMs-SrNH catalysts.** Surface reaction profiles for (**a**) Fe/SrNH, (**b**) Co/SrNH and (**c**) Ni/SrNH catalysts with the reaction gas of N₂ and D₂ at the temperature increased from room temperature to 400 °C. Prior to N₂/D₂ isotopic reaction, each sample is pretreated in H₂ + N₂ at 400 °C for 24 h. **d** Reaction rate of N₂ isotope exchange over Fe/SrNH, Co/SrNH and Ni/SrNH catalysts at 26.7 kPa (¹⁵N₂:¹⁴N₂ = 1:4). Each sample is pretreated in H₂ at 400 °C for 24 h before the ¹⁴N/¹⁵N isotopic experiment. Error bars represent the standard deviation from three independent measurements.

that similar H₂-TPR data is also confirmed for non-loaded SrNH (Fig. 4c, d), evincing that the peak at ca. 350 °C is indeed derived from the reaction between H* and the lattice NH²⁻ of SrNH. The greater amount of the desorbed ammonia in Ni/SrNH suggests that the enhanced NH₃ formation is caused by the aid of dissociated H* from Ni. Different from Fe/SrNH and Ni/SrNH, Co/SrNH can be regarded as "hybrid" mechanism because of moderated interaction between Co and N. As shown in Fig. 4b, the Co/SrNH catalyst shows two major NH₃ desorption peaks at ca. 200 °C and ca. 350 °C, which indicates the adsorption and activation of reactant N₂ are associated with both loaded Co metal and NH²⁻ vacancy sites of SrNH support. It was thus demonstrated that the synergy of TMs and NH²⁻ vacancies plays a critical role for ammonia production, and multiple reaction pathways should be realized over Fe/SrNH, Co/SrNH and Ni/SrNH, respectively.

To investigate the synergistic functionalities between TMs and NH²⁻ vacancies in TMs/SrNH catalysts, DFT calculations and detailed experimental characterizations were conducted. Bader charge analysis showed the three TMs are negatively charged, i.e., −0.08 (Fe), −0.14 (Co), and −0.10 (Ni) (Fig. S16). Co accepts more electrons than Fe because of its deeper 3d orbitals. Meanwhile, Ni accept fewer electrons than Co because its 3d orbitals are almost occupied. It is consistent with the XPS results, in which these 2p peaks of TMs shifted to lower binding energy region (Fig. S17). Such negatively charged TMs can be ascribed to the electron transfer from the NH²⁻ vacancy of SrNH to TMs (Fig. S18). Next, we evaluated how the NH²⁻ vacancy affects the

electron donation ability of SrNH. Figure 5a shows the calculated density of states of SrNH with and without NH²⁻ vacancy. Compared with SrNH, an anionic electron state appeared between the valence and conduction band (Fig. S19), which is derived from the confined electron at the NH²⁻ vacancy of SrNH. Accordingly, the calculated work function (Φ_WF) for NH²⁻ vacancy containing SrNH (SrNH_VNH, Φ_WF ~ 2.0 eV, Fig. 5b) become smaller than that for defect-free SrNH (Φ_WF ~ 2.6 eV) (Fig. S20), showing that its electron donation ability is further strengthened by generating NH²⁻ vacancy. We thus considered that the confined anionic electrons can be donated effectively to TMs (Φ_WF ~ 5.0 eV) and thus facilitate N₂ dissociation through the back-donation of electrons to anti-bonding π* orbitals of N₂, and this electron transfer mechanism is particularly effective for Fe/SrNH and Co/SrNH.

To confirm the formation of the NH²⁻ vacancies, AES has been conducted to investigate the surface composition of fresh and H₂-treated SrNH samples (Fig. 5c). The normalized intensity of N peak (ca. 387 eV) decreased clearly after the H₂ treatment process, suggesting the reaction of surface NH²⁻ species transformed to NH₃ by reacting with H*. The generation of surface NH²⁻ vacancies was further confirmed by high-resolution transmission electron microscopy (HR-TEM). Since the light element of N and H gave low contrast in HRTEM, the lattice Sr of SrNH support was detected with a hexagonal pattern along (111) direction (Fig. 5d). With the H₂ treatment, a distortion of the hexagonal pattern of lattice Sr could be identified

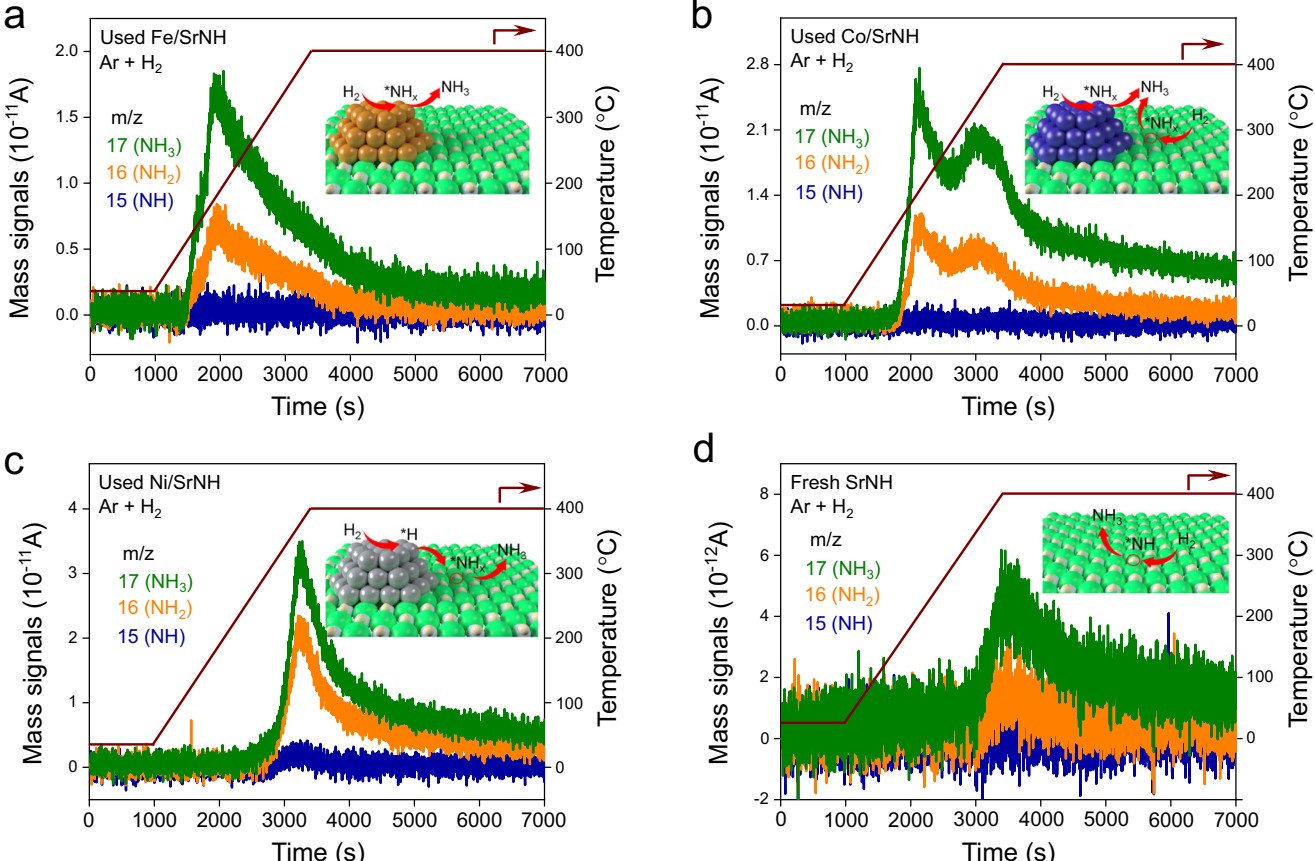

**Fig. 4 | H$_2$-Temperature programmed reaction (TPR) of TMs-SrNH catalysts.** H$_2$-TPR profiles for used (**a**) Fe/SrNH, (**b**) Co/SrNH, (**c**) Ni/SrNH, and fresh (**d**) SrNH catalysts under Ar and H$_2$ (20 mL min$^{-1}$ reaction gas, Ar/H$_2$ = 1:1) at the temperature increased from room temperature to 400 °C. Prior to H$_2$-TPR measurements, the samples denoted as used catalysts (**a**–**c**) were treated at 400 °C for 24 h under the NH$_3$ synthesis condition (60 mL min$^{-1}$ reaction gas, N$_2$/H$_2$ = 1:3).

along the same direction (Fig. 5e), consistent with the formation of a substantial amount of surface NH$^{2-}$ vacancies. The E$_V$ are calculated as Co/SrNH (1.59 eV) < Ni/SrNH (1.75 eV) < Fe/SrNH (1.84 eV) < SrNH (1.92 eV) (Fig. 5f), indicating that the formation of NH$^{2-}$ vacancies becomes easier upon TM-loading. Accordingly, the observed intensity change of N AES peak shows that the surface NH$^{2-}$ was consumed by H$_2$ and NH$^{2-}$ vacancies are likely to be generated on SrNH surface (Fig. S21). To further strengthen our claim, we further performed X-ray photoelectron spectroscopy (XPS) characterizations to illustrate the presence of surface NH$^{2-}$ deficiencies. As shown in Fig. S22, N 1$s$ XPS peaks of fresh Co/SrNH are located at around 400 eV and the intensity of this N peak is significantly weakened after H$_2$ treatment. After etching the surface by argon plasma, it is found that the N peak intensity was recovered and became the same level as that of fresh Co/SrNH. At the same time, the intensity of Sr XPS peaks is largely no changed. These results indicated a substantial amount of NH$^{2-}$ vacancies are formed on SrNH surface after H$_2$ treatment.

The NH$^{2-}$ vacancies could be filled by other species such as H$^-$ anions. To examine this, we also performed a H$_2$-TPR and Ar-TPD experiment. In H$_2$-TPR measurement, 0.1 g of fresh Co/SrNH was treated under pure H$_2$ atmosphere at 400 °C for 24 h, and ~0.075 mmol NH$_3$ could be detected (Fig. S23a). It means that 0.075 mmol lattice NH$^{2-}$ was consumed during the H$_2$ treatment. Subsequently, Ar-TPD measurements were conducted to estimate the incorporated H$^-$ ions and the amount of the desorbed H$_2$ was estimated to be 0.025 mmol (i.e., 0.05 mmol H$^-$ ions, Fig. S23b), which indicates significant amount of NH$^{2-}$ vacancies are occupied by H$^-$ ions. DFT calculation also demonstrated that the NH$^{2-}$ vacancy is favored to capture H$^-$ as anions (Table S7). We acknowledge that H$^-$ ions can be accommodated in partial NH$^{2-}$ vacancy sites, which is also beneficial to the reduction of N$_2$ to promote ammonia synthesis[37]. Meanwhile, it should be noted the desorbed H$^-$ ions amount (0.05 mmol) was smaller than that of consumed lattice NH$^{2-}$ (0.075 mmol), which demonstrate the existence of NH$^{2-}$ vacancy in Co/SrNH. Such high amount of incorporated H$^-$ ions would not lead to a change of the surface structure of SrNH (Fig. S24). XPS measurement shows that the valence states of the Sr species of H$_2$-treated Co/SrNH are similar to that of fresh one (Fig. S22), suggesting the stable surface structure of SrNH during the H$_2$ treatment. Most importantly, in Raman spectra, the absence of hydrogen vibration (broad bands at the wavelength of 400−1000 cm$^{-1}$)[61,62] further exclude the surface generation of SrH$_2$ in H$_2$-treated Co/SrNH (Fig. S25).

The energy profiles for lattice NH$^{2-}$ hydrogenation over SrNH and N$_2$ dissociation on TMs were investigated using DFT. As shown in Fig. 6a, the hydrogenation steps of the generated NH$_2^-$ species (TS2) gave the highest energy states among all the reaction coordination for each catalyst. It was detected that the energy barrier for NH$_3$ formation on Fe/SrNH (1.66 eV) is much higher than those on Co/SrNH (1.26 eV) and Ni/SrNH (1.37 eV), reflecting the energetically unfavored pathway through the lattice NH$^{2-}$ of the SrNH support over Fe catalyst (Tables S8–S9, Figs. S26–S28). Meanwhile, the comparable values over Co/SrNH and Ni/SrNH demonstrate that NH$_3$ formation through the SrNH support are both energetically preferred. The slightly lower barrier of Co/SrNH indicates that the lattice NH$^{2-}$ in Co/SrNH is more easily hydrogenated by H* than that in Ni/SrNH, which is consistent with the difference of calculated E$_V$ (Fig. 5f).

Subsequently, the N$_2$ dissociation on the surface of TMs were investigated. The E$_a$s of TM/SrNH were calculated to be 1.19 eV (Fe),

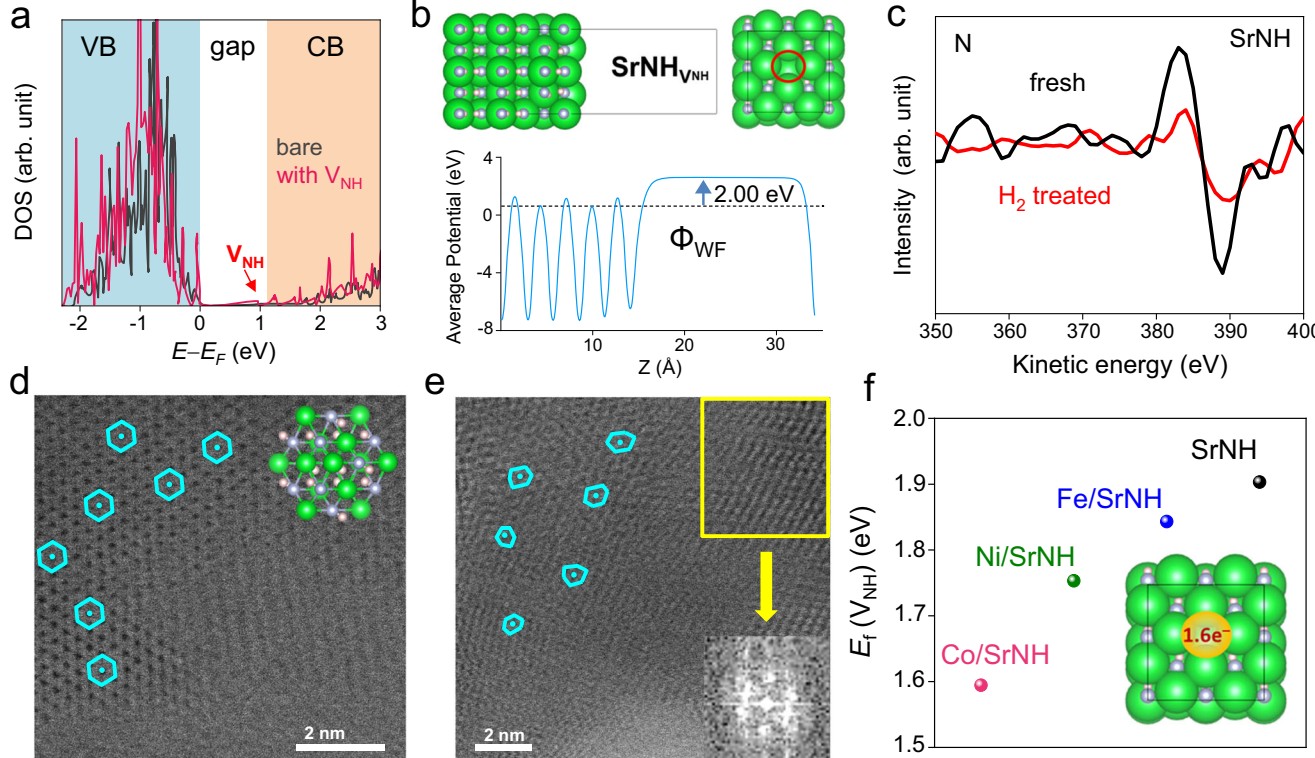

**Fig. 5 | Calculation and characterizations of $V_{NH}$ of TMs-SrNH catalysts.**
**a** Projected density of states (DOS) for SrNH with and without $V_{NH}$. **b** Calculated work function of SrNH with surface $NH^{2-}$ vacancy. **c** AES spectra for N, Sr and TMs of fresh and $H_2$-treated SrNH. HRTEM image of (**d**) fresh and (**e**) $H_2$-treated SrNH along the [111] direction. The inset of panel (**d**) shows the crystal structure of SrNH along the [111] direction. Sr, N and H atoms are represented as green, gray and light pink balls, respectively. The inset of (**e**) shows the corresponding FFT pattern of the yellow region. **f** $V_{NH}$ formation energies $E_V$ over bare SrNH and various TMs-SrNH catalysts. Inset shows the electron density in the region of $V_{NH}$ site of bare SrNH.

1.35 eV (Co) and 2.22 eV (Ni), respectively (Fig. 6b, Fig. S29, Table S10). Here, we define the difference of the maximum energy barrier ($\Delta E_a$) between the TMs and SrNH support pathways as a rough descriptor of anticipated activity along each pathway (Fig. 6c), in which positive values of $\Delta E_a$ indicate that the SrNH support route is favored while negative values indicate TMs play more important role for ammonia synthesis. Fe/SrNH shows a $\Delta E_a$ value of −0.47 eV, suggesting that nitrogen and hydrogen reacted with each other majorly on Fe metal surface to produce $NH_3$ (Fig. 6d). This is in good agreement with the $^{15}N_2/H_2$ isotopic experimental results, in which lattice $NH^{2-}$ of SrNH support is not involved in ammonia product (Fig. 2a). In the case of Co/SrNH, the nearly zero value (0.09 eV) of $\Delta E_a$ implies the comparable energy barrier for both reaction pathways, i.e., both Co metal and SrNH support contributed to the formation of ammonia product, which can be confirmed by $H_2$-TPR results as well (Fig. 4b). Remarkably, the comparable overall energy barrier of -0.6 eV (ca. 57.9 kJ·mol⁻¹) for SrNH support route (Fig. 6a) and -0.5 eV (ca. 48.2 kJ·mol⁻¹) for Co metal route (Fig. 6b) are also close to the experimentally obtained $E_a$ from the Arrhenius plot (ca. 55 kJ·mol⁻¹) (Fig. 1b). Therefore, we proposed a dual reaction pathway through the synergy of TMs and SrNH in Co/SrNH (Fig. 6e), in which the small work function (~2.0 eV) of SrNH support accounts for the strong electron donation ability that facilitates $N_2$ activation on Co metal. Meanwhile, the formed $NH^{2-}$ vacancy of the SrNH support provides additional active sites to the adsorption of $N_2$ and then hydrogenation to $NH_3$. As for Ni/SrNH catalyst, the weak interaction between Ni and N gave rise to the highest energy barrier for $N_2$ dissociation as well as a positive value of $\Delta E_a$. Instead, $N_2$ molecules are adsorbed and activated at the in situ formed $NH^{2-}$ vacancy sites, and continuously react with H* from Ni, to realize a stable catalytic

ammonia synthesis cycle over SrNH support (Fig. 6f), similar to the previously studied reported Ni/ReN system. The rate-determining step for the ammonia formation is associated with the combination of H* and $NH^{2-}$ with a calculated energy barrier of 1.37 eV, and the overall energy barrier for $NH_3$ formation through the hydrogenation process is roughly 0.8 eV (ca. 77.2 kJ·mol⁻¹), comparable to the aforementioned $E_a$ (ca. 90 kJ·mol⁻¹) of Ni/SrNH (Figs. 1b and 6a). Such high energy barrier should hinder the diffusion of H* from Ni to SrNH in some degree, leading to a high H coverage over Ni catalysts, which accounts for almost zero $H_2$ reaction order (β) (Table 1).

## Discussion

In our previously investigated ReN catalyst systems, the nitrogen vacancy not only provides active sites for $N_2$ activation and hydrogenation to form $NH_3$, but also reduces the work function of the ReN support, facilitating the $N_2$ dissociation on loaded TMs metal. In this study, since the much larger anionic vacancy size of SrNH are realized compared to ReN, the energy barrier of lattice $NH_x$ hydrogenation was significantly reduced over SrNH-based catalysts probably due to the steric effect (Fig. S30). Accordingly, the removal of lattice $NH^{2-}$ from Ni/SrNH by $H_2$ proceeds more easily than from Ni/CeN (Fig. S31). By employing $NH^{2-}$ vacancy, the ammonia production over SrNH support became more efficient than that of ReN. On the other hand, a low work function is also a highlighted feature in SrNH with $NH^{2-}$ vacancy, which promotes the activation of $N_2$ molecules. Compared with the work function ($\Phi_{WF}$ = ~2.3 eV) of $ReN_V$, a much lower work function ($\Phi_{WF}$ = ~2.0 eV) was realized by $V_{NH}$ formation on the SrNH support (Fig. S32), enabling much stronger electron donation from SrNH to Co that can facilitate $N_2$ cleavage on Co, which leads to significantly enhanced catalytic activity of Co metal for ammonia synthesis. Overall, an

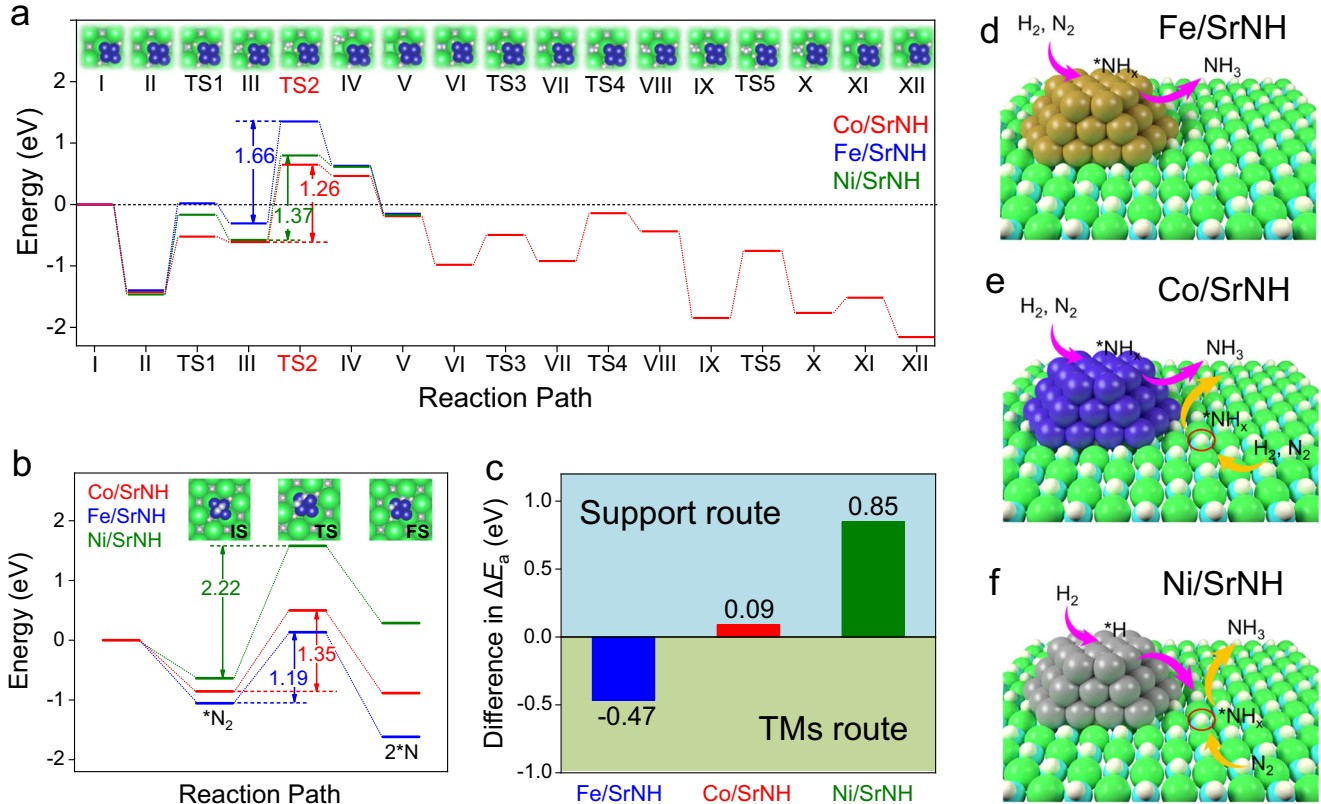

**Fig. 6 | Theoretical calculation of TMs-SrNH catalysts. a** Calculated energy profiles of support route for $N_2$ activation and hydrogenation at the $V_{NH}$ site of SrNH support (Support route) over Fe/SrNH, Co/SrNH and Ni/SrNH catalysts. Inset shows the structures of the intermediates and transition states (TSs) for the key elementary steps over Co/SrNH. **b** Calculated energy profiles of $N_2$ activation on the surface of TMs (TMs route) in TMs-SrNH catalysts. **c** Calculated difference in energy barrier between the TMs and support reaction pathways for ammonia synthesis process on Fe-, Co- and Ni-loaded SrNH catalysts. $\Delta E_a$ is described as $E_a$ (TM Route) $-E_a$ (Support route). Proposed reaction pathway for ammonia synthesis over (**d**) Fe/SrNH, (**e**) Co/SrNH and (**f**) Ni/SrNH catalysts.

unprecedented high reaction rates were achieved on Co/SrNH as shown in Figs. S13, S14 and Table S1–S3. Therefore, the generated $V_{NH}$ plays a dominant role during the reaction, and both Co and $V_{NH}$ on SrNH served as the active centers for $N_2$ activation and ammonia production.

The present work demonstrates that $Ae$NH can act as efficient supports for promoting various TMs catalysts in ammonia synthesis. The synergy of TMs and in situ formed $NH^{2-}$ vacancy of $Ae$NH has a decisive effect on the reaction pathway and thus result in distinct catalytic performance. In Fe and Ni cases, the ammonia formation is separately realized at Fe metal and $V_{NH}$ sites of the support, respectively. While, a combination of TMs and $V_{NH}$ route can be achieved over Co/SrNH catalyst, which was proved to be the most efficient catalyst for ammonia synthesis among the different TMs/$Ae$NH catalysts investigated. The in situ generated $V_{NH}$ of SrNH not only supplies surface active sites for $N_2$ activation and hydrogenation to $NH_3$, but also reduces the work function of SrNH support, promoting $N_2$ dissociation as well as $NH_3$ formation on Co metal. As a results, the catalytic activity of Co/SrNH far exceed the other reported Co- and Ni-based catalysts, and even higher than conventional Ru-based catalysts and industrial Fe-based catalyst. The present findings provide important information toward understanding the synergy effect of TMs and $Ae$NH support on the reaction pathway for ammonia synthesis.

## Methods
### Sample preparation
$Ae$H$_2$ were prepared by the reaction of alkaline earth metal ingot (99.99% purity) with $H_2$ gas by an Ar/$H_2$ arc evaporation system[23]. In the arc evaporation process, the Ar and $H_2$ partial pressures were set to

0.04 MPa and 0.01 MPa respectively, and the reaction current was set to 60 - 80 A. Subsequently, $Ae$NH were synthesized by reacting $Ae$H$_2$ NPs under $N_2$ atmosphere at 400 °C for 48 h. Iron carbonyl [$Fe_2(CO)_9$], cobalt carbonyl [$Co_2(CO)_8$] and nickelocene [$Ni(C_5H_5)_2$] were used as TMs precursors respectively. Then each TMs precursor and $Ae$NH were mixed by hand-mill in agate mortar. The mixture was then heated in pure $H_2$ flow to produce TMs-$Ae$NH. Since $Ae$NH is moisture sensitive, all of the preparation procedures were performed in the Ar-filled glovebox. The illustration of the preparation process and corresponding powder XRD patterns of the as prepared $Ae$H$_2$ and $Ae$NH are shown in Fig. S33.

Other reference support materials, such as Ba-Ca(NH$_2$)$_2$, CeN, and C12A7:e$^-$ electride were prepared according to our previously reported method[26,32,47], whereas MgO and SrO were commercially available products. Co-loading was conducted according to the same thermal reduction process as that used for SrNH. LaCoSi was fabricated by arc-melting process using stoichiometric amounts of lanthanum, cobalt, and silicon ingots. The obtained ingot was annealed at 1000 °C while wrapped in a sealed quartz tube for 5 days, and then purified by further annealing at 800 °C for 10 days. The preparation of Co$_3$Mo$_3$N was realized through a nitridation process by using CoMoO$_4$ as precursor. CoMoO$_4$ was heated in a quartz reactor under $NH_3$ gas flow at 800 °C for 5 h. Before taking out the sample from the reactor, pure $N_2$ was used to purge residual $NH_3$ in the reactor. For the preparation of Cs-Ru/MgO, MgO was treated in high vacuum at 500 °C for 6 h and then mixed with Ru$_3$(CO)$_{12}$ in the Ar-filled glovebox. The obtained powder was sealed in a quartz tube and slowly heated to 250 °C for 2 h. The obtained dark gray powder was dispersed in an absolute ethanol solution of Cs$_2$CO$_3$ by stirring for 3 h, and then the solvent was

removed by evaporation and the catalyst was dried in vacuum. The atomic ratio of Cs/Ru in the catalyst was 1.0 and the Ru content was determined to be 10.0 wt%.

## Catalytic reaction

Catalytic reactions were conducted in a fixed-bed flow system. In a typical run, 0.1 g catalyst was pretreated in a stream of $N_2$:$H_2 = 1$:3 under WHSV of 36,000 $mL \cdot g^{-1} \cdot h^{-1}$ and at 0.1 MPa using a temperature program of heating to 400 °C for 1 h and then holding at 400 °C for 2 h. The ammonia produced was monitored under steady-state conditions of temperature (250 - 400 °C) with a flow rate of 60 $mL \cdot min^{-1}$ at 0.1 - 0.9 MPa. The ammonia produced was trapped in 5 mM sulfuric acid solution and the amount of $NH_4^+$ generated in the solution was determined using ion chromatography (Prominence, Shimadzu) with an electrical conductivity detector. Comparison of the catalyst performance was conducted under the same conditions.

Temperature-programmed reduction with $H_2$ ($H_2$-TPR) was also conducted in a fixed-bed flow system. 0.1 g catalyst was treated in a pure $H_2$ flow (60 $mL \cdot min^{-1}$) at 0.1 MPa using a temperature program of heating (3 $°C \cdot min^{-1}$) to 400 °C and then holding at 400 °C for 24 h. The produced ammonia was dissolved in 5 mM sulfuric acid solution and the amount of $NH^{4+}$ ions in the solution was identified using the same instrument for ammonia synthesis. Ar-Temperature-programmed desorption (Ar-TPD) (BELCAT-A, BEL) was also performed. Prior to measurements, 0.1 g catalyst was introduced into a quartz glass cell in an Ar-filled glovebox and the glass cell was heated (10 $°C \cdot min^{-1}$) in an Ar stream (50 $mL \cdot min^{-1}$), and the concentration of $H_2$ was monitored with a thermal conductivity detector (TCD) and mass spectrometer (Bell Mass, BEL).

Since the active sites of Co/SrNH are consist of both transition metal and surface $NH^{2-}$ vacancy, the calculation of turnover frequencies (TOFs) is based on the total amount of surface active sites including surface Co metal atoms and surface NH vacancy sites.

The amount of surface Co sites are derived from the average particle size observed by TEM. Assuming that the Co particles are semispherical, the TOF based on surface Co is calculated below:

The Co weight $W_M$ [g] is calculated as Eq. (1):

$$W_M = m \times \frac{c}{100} \tag{1}$$

where m [g] is the weight of the catalyst and c [%] is Co loading weight percentage.

The specific surface area of Co, $A_M$ [$m^2 g^{-1}$], is calculated as Eq. (2):

$$A_M = \frac{0.5 \times 4\pi \left(\frac{d}{2} \times 10^{-9}\right)^2 \times a}{W_M} \tag{2}$$

where d [nm] is the average diameter of the Co particles, a [count] is the number of Co particles, and $W_M$ [g] is the weight of the Co.

The specific volume of Co, $V_{Co}$ [$m^3 g^{-1}$], is calculated as Eq. (3):

$$V_M = \frac{0.5 \times \frac{4}{3}\pi \left(\frac{d}{2} \times 10^{-9}\right)^3 \times a}{W_M} = \frac{1}{\rho \times 10^6} \tag{3}$$

where $\rho$ [$g cm^{-3}$] is the density of Co, which is 8.9 [$g cm^{-3}$].

Based on Eqs. 2 and 3, $A_M$ is solved as Eq. (4):

$$A_M = \frac{6000}{d \times \rho} \tag{4}$$

The number of Co surface sites $N_S$ [count] is calculated as Eq. (5):

$$N_S = \frac{A_M \times W_M}{S_M \times 10^{-18}} \tag{5}$$

where $S_M$ [$nm^2$] is the cross-sectional area per Co atom, which is 0.066 [$nm^2$]. Here, since the average particle size of Co metal of Co (1.5 wt%)/SrNH and Co (4.8 wt%)/SrNH is 11 nm and 17 nm (Fig. S12), respectively, the estimated number of Ns is $1.4 \times 10^{18}$ and $2.8 \times 10^{18}$ for the Co (1.5 wt%) and Co (4.8 wt%).

The amount of surface vacancy sites was calculated from the concentration of top-layer $NH_{lattice}$ of imide. From the unit cell of pure SrNH, the amount of top-layer $NH_{lattice}$ was estimated as Eq. (6):

$$N_{NH} = \frac{1}{a^2} \times S_{BET} \times g_{cat} \tag{6}$$

where a is the lattice parameter of SrNH (a = $5.64 \times 10^{-10}$ m), $S_{BET}$ is the surface area of Co/SrNH ($S_{BET}$ = 15.9 $m^2 \cdot g^{-1}$), and $g_{cat}$ (=0.1 g) is the amount of catalyst used during the experiment. According to the lattice structure, one NH site presents per the area of $3.18 \times 10^{-19}$ $m^2$. Therefore $n_{NH} \sim 5.03 \times 10^{18}$ of NH sites exist in the used sample according to the lattice structure of SrNH. According to our experimental results in Figure S23, around 2/3 of the NH defects are occupied by $H^-$, which indicates that 1/3 of the surface NH vacancies can serve as the catalytic active sites. Thus, the amount of surface vacancy active sites should be $1.7 \times 10^{18}$.

According to above calculation of the amount of surface active sites, the TOF [$h^{-1}$] is calculated as Eq. 7:

$$TOF = \frac{r_{NH_3}}{N_S + N_{NH}} \times 10^{-6} \times 6.02 \times 10^{23} \tag{7}$$

where $r_{NH_3}$ [$\mu mol\ g^{-1}\ h^{-1}$] is the ammonia synthesis rate.

## Kinetic analysis

The apparent $E_a$ were calculated from Arrhenius plots for the reaction rates, which were <20% of that at equilibrium. Measurement of the reaction orders for $N_2$ and $H_2$ was conducted with Ar gas as a diluent to ensure a total flow of 60 mL $min^{-1}$ when changing the flow rate of $N_2$ and $H_2$. The reaction orders were estimated by using the following Eqs:

$$r = k \times P_{N_2}^{\alpha} \times P_{H_2}^{\beta} \times P_{NH_3}^{\gamma} \tag{8}$$

$$r = \frac{1}{W} \frac{dy_0}{d\frac{1}{q}} \tag{9}$$

$$\log y_0 = \log \left(\frac{C}{q}\right)^{\frac{1}{m}} \tag{10}$$

$$r = \frac{1}{W} \times \frac{C}{m} \times y_0^{1-m} \tag{11}$$

$$C = k_2 \times P_{N_2}^{\alpha} \times P_{H_2}^{\beta} \tag{12}$$

where r, W, $y_0$, q, and (1-m) represent the reaction rate of the ammonia synthesis, the catalyst weight, the mole fraction of $NH_3$ at the reactor outlet, the flow rate, and the reaction order with respect to $NH_3$ ($\gamma$). Finally, the $\alpha$ and $\beta$ can be determined by plotting the logarithm of "C" vs that of $N_2$ or $H_2$ partial pressure.

## Isotopic experiments

Ammonia synthesis was performed in a U-shaped glass reactor connected to a closed gas circulation system. The reactants, $^{15}N_2$ (98%) and $H_2$ with a ratio of 1:3, were introduced into the system with a total pressure of 60 kPa and then heated to 400 °C. The composition of the circulating gas through the system was detected by utilizing a quadrupole mass spectrometer (M-101QA-TDM, Canon Anelva Corp.), using Ar as the carrier gas. To overcome the limitations of reactant gas diffusion and adsorption/desorption, a circulating pump was introduced into the system. Masses with m/z values of 2, 16, 17, 18, 28, 29, and 30 were monitored over time to track the progress of the reaction.

The experiment of $N_2/D_2$ and $H_2$-Temperature-programmed reaction ($H_2$-TPR) was carried out in a fixed-bed flow system. Both $N_2/D_2$ and $Ar/H_2$ gases at a flow ratio of 1 were used with a total flow rate of 20 mL·min$^{-1}$. The fixed-bed reactor was then heated to 400 °C at a rate of 10 °C/min and 0.1 MPa. To analyze the reaction products, an online mass spectrometer (ANELVA, Quadrupole Mass Spectrometer) was used. The masses $m/z = 16, 17, 18, 19,$ and 20 were monitored over time to track the progress of the reaction.

The $N_2$ isotope exchange experiment was conducted in a closed gas circulation system equipped with a U-shaped glass reactor. The reactants, a mixture of $^{15}N_2$ and $^{14}N_2$ in a 1:4 ratio, were introduced into the system at a total pressure of 20 kPa. Afterward, the mixture was heated to 400 °C until reaching adsorption equilibrium. The gas in circulation was continuously monitored using a quadrupole mass spectrometer (M-101QA-TDM, Canon Anelva Corp.), measuring the mass-to-charge ratios ($m/z$) of 28, 29, and 30 as a function of time.

## Sample characterization

The crystal structure was analyzed using XRD (D8 Advance, Bruker) with Cu Kα radiation ($\lambda = 0.15418$ nm). The sample was put in an X-ray transmitting capsule to protect it from oxidation. High-resolution transmission electron microscopy (HR-TEM) images were obtained using a JEOL JEM-ARM300F atomic resolution analytical electron microscope operated at an accelerating voltage of 300 kV. X-ray photoelectron spectroscopy (XPS; ESCA-3200; Shimadzu) measurements were performed using Mg Kα radiation at <10$^{-6}$ Pa (8 kV bias voltage applied to the X-ray source). XPS data were corrected according to the C (carbon) 1 s peak (binding energy = 284.6 eV). Auger electron spectra were obtained with 10 keV primary electrons using a scanning Auger nanoprobe system (PHI 710, Ulvac-Phi). Raman spectra were measured with a spectrometer (HR-800, Horiba Jobin Yvon), using a laser with a wavelength of 457 nm. Nitrogen sorption measurements (BELSORP-mini II, BEL, Japan) were applied to evaluate the Brunauer-Emmett-Teller surface areas of the catalysts.

## Theoretical calculations

All DFT calculations were got through the Vienna ab initio simulation package (VASP)[63]. The management of the electron exchange and correlation energy was generalized gradient approximation method with the Perdew–Burke–Ernzerhof (PBE) exchange–correlation functional[64], while the projector augmented wave (PAW) method[65,66] was employed to describe the core electrons. The description of valence electrons was using plane wave basis kinetic energy cut-off value with 450 eV. A k-point mesh of $2 \times 2 \times 1$ was used from the Gamma center. Meanwhile, in the case of system, the unit cell was $16.82 \times 16.82 \times 39$ Å$^3$, the thickness of the vacuum layer was 25 Å. The upper two layers of each slab were allowed to relax (Fig. S34), while the bottom layers were constrained to their original positions. The 6-atoms metal cluster with octahedron structure was loaded on slab before relaxation. All models were fully optimized until the energy and forces are converged to $1 \times 10^{-5}$ eV and 0.0257 eV Å$^{-1}$, respectively. The $Ae$NH (001), based on the surface energy of the three facets with low miller index, was selected for study because it was the most stable facet (Table S11). To introduce TM cluster, we chose six Co atoms model with an octahedron structure since 1-4-1 is the most stable 3D configuration[67]. Then, two sites were considered for the cluster location. Table S12 shows the total energy of Co cluster loaded on NH site and Sr site of SrNH (001) facet respectively. It is clear that NH site loading is more energy favored. For TS calculation, the parameters of CI-NEB[68] were kept the same as the structure relaxation. Three transition states were interpolated linearly between initial state and final state by VASPKIT package[69].

The formation energy of NH vacancy was conducted following Eqs. (13) and (14):

$$\Delta E_V = E_V + E_{NH} - E_{slab} \tag{13}$$

$$E_{NH} = (E_{nitrogen} + E_{hydrogen})/2 \tag{14}$$

Meanwhile, the formation energy of NH vacancy with the help of H was also conducted by $E_{NH} = E_{NH3} - E_{H2}$. And the tendency of the vacancy formation energy with respect to $NH_3 - H_2$ is almost the same to those of $N_2$ and $H_2$ (Table S13).

All the binding energies of intermediates were $E = E_X - E_{slab} - E_{*X}$ based on following Eq. (15):

$$X + slab \rightarrow {}^*X \tag{15}$$

while X was the intermediates during ammonia synthesis, and *X was the intermediates adsorbed on catalyst.

## Data availability

The data generated in this study are presented in the main text and Supplementary Information, and can be obtained from the corresponding authors upon reasonable request.

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

## Acknowledgements

This work was supported by the National Natural Science Foundation of China (22275121, 22105122), the Science and Technology Commission of Shanghai Municipality (21PJ1407400), Shanghai Municipal Science and Technology Major Project, Shanghai Science and Technology Plan (21DZ2260400). The authors also thank the support from the Open Foundation Commission of Shaoxing Research Institute of Renewable Energy and Molecular Engineering (JDSX2022038), the project of Jiangxi Academy of Sciences (2023YSTZX01), C$\hbar$EM (02161943), SPST, Shanghai Tech University. A part of this work was also supported by the project (JPNP21012) commissioned by the New Energy and Industrial Technology Development Organization (NEDO), JST FOREST Program (JPMJFR203A), and Kakenhi Grants-in-Aid (JP22H00272, JP21H00019) from the Japan Society for the Promotion of Science (JSPS). Y.F.L. was financially supported by the start-up fund from Chongqing University (02110011044171).

## Author contributions

T.-N.Y. conceived the idea. T.-N.Y., M.K., and H.H. supervised the project. Z.L., T.-N.Y., Y.L., J.L., M.X., and M.K. performed the synthesis, characterization, and catalytic measurements. Z.L. and S.-W.P. conducted the model construction and DFT calculations. Y.Q. helped with the STEM measurements. T.-N.Y., Z.L., Y.L, J.-S.C., and H.H. co-wrote the paper with input from all authors.

## Competing interests

The authors declare no competing interests.
