## [Peer Review File · Nature Communications]

REVIEWER COMMENTS

Reviewer #1 (Remarks to the Author):

As I indicated in my previous report, this highly work-loaded contribution is interesting and worth of publication in Nature Comm., albeit I still cannot be convinced by the authors on the novelty issue. I noticed the same group have investigated $\text{Ba}(\text{NH}_2)_2$, SrNH , Ca_2NH , CaNH etc. as catalyst supports for ammonia synthesis and decomposition, NH (or N) vacancies were found important for the catalysis. It would be interesting to know whether Ca_2NH and CaNH behave differently.

The authors showed H_2 -TPR and Ar-TPD data evidencing H_2 desorption from the hydrogen treated sample, and pointed out that H may occupy the NH vacancies. If so, there could be no vacancies but composition change to the support, especially on the surface. Their XPS data also evidence considerable compositional change. I'd suggest the authors to use proper term to describe their catalysts.

It is not a good idea to compare TOF among literature reports unless active site is well defined. There are a number of ways people calculate TOF. Reporting activity data based on the mass of catalyst is better.

Response Letter

Response to the referees

Referees' comments (gray)

Our responses (black)

Reviewer #1 (Remarks to the Author):

As I indicated in my previous report, this highly work-loaded contribution is interesting and worth of publication in Nature Comm., albeit I still cannot be convinced by the authors on the novelty issue. I noticed the same group have investigated Ba(NH₂)₂, SrNH, Ca₂NH, CaNH etc. as catalyst supports for ammonia synthesis and decomposition, NH (or N) vacancies were found important for the catalysis. It would be interesting to know whether Ca₂NH and CaNH behave differently.

Response: We appreciate the referee for this comment. The major difference for Ca₂NH and CaNH is the charge states of hydrogen, i.e., hydrogen is negatively charged (H⁻) for Ca₂NH, while it becomes a cation (H⁺) in CaNH, which significantly modifies its physical properties. Ca₂NH can be regarded as the hydride of Ca₂N electride in which H⁻ ions are inserted between the two-dimensional Ca₂N layers. In Ca₂NH, hydrogen interacts with Ca. Meanwhile, in CaNH, hydrogen is bonded with N with a positive charge, forming the (NH)²⁻ polyanions.

Ca₂NH has a low work function and reversible hydrogen-electron exchange reaction under ammonia synthesis conditions, which facilitates ammonia synthesis (*Chem. Sci.*, 2016, 7, 4036), while CaNH is a semiconductor and has a much higher work function compared to Ca₂NH. Consequently, Ru/Ca₂NH showed high ammonia synthesis activity with a low activation energy of 60 kJ/mol, but Ru/CaNH showed poor activity with an activation energy of 110 kJ/mol.

In contrast to the previous studies, the present work focused on the functionalities of (NH)²⁻ polyanionic defects that offer more catalytically active sites for N₂ adsorption and activation. The presented data demonstrate that the large polyanionic defects in combination with transition metals (Fe, Co, Ni) offer controllable N₂ activation processes.

In summary, while the chemical formulas are similar between Ca₂NH and CaNH, the structure, properties and catalytic mechanism are totally different. The Ca₂NH is based on the electron donation ability of Ca₂NH, and it hardly activate N₂ without the help of Ru. Ni/Ca₂NH is thus less effective for ammonia synthesis. The CaNH utilizes its defect structure, highlighting its

advantages particularly by combining with the ubiquitous elements-based catalysts, such as Co and Ni. This fact offers a flexible material design for ammonia synthesis.

The authors showed H₂-TPR and Ar-TPD data evidencing H₂ desorption from the hydrogen treated sample, and pointed out that H may occupy the NH vacancies. If so, there could be no vacancies but composition change to the support, especially on the surface. Their XPS data also evidence considerable compositional change. I'd suggest the authors to use proper term to describe their catalysts.

Response: We appreciate the reviewer's comment. From the XPS results (Figure R1), it is noted that only the intensity of the N peaks decreased after H₂ treatment. We attributed this "composition change" to the transformation of surface (NH)²⁻ species to NH₃ by reacting with H*, i.e., the formation of (NH)²⁻ vacancies. It is consistent with the HRTEM result, in which the distortion of the lattice are confirmed due to the generation of a substantial amount of surface (NH)²⁻ vacancies. We further etch the surface of this H₂-treated sample by argon plasma, the N peaks intensity would be recovered and became the same level as that of fresh Co/SrNH, indicating the NH vacancies are existed at the surface of the catalyst. We acknowledge that hydridic hydrogen can be accommodated in partial NH²⁻ vacancy sites. It is reported that hydridic hydrogen is a combined source of electrons and protons, which was found to exert profound effects on nitrogen fixation (Chem. Soc. Rev. 2014, 43, 547–564). Thus, we consider the surface incorporated H⁻ anions should be dynamically consumed and participated in the reduction of N₂ to promote ammonia synthesis.

Figure R1. XPS spectra of (a) Sr 3d and (b) N 1s in fresh Co/SrNH, H₂ treated Co/SrNH and with Ar plasma etching.

Figure R2. Raman spectra of SrH₂ and H₂ treated Co/SrNH.

On the other hand, Raman spectra (Figure R2), further excluded the possibility of surface structure modification. For example, the absence of hydrogen vibration (broad bands at the wavelength of 400-1000 cm⁻¹) excluded the generation of SrH₂ in H₂-treated Co/SrNH. Combining the XPS and Raman results, we consider the surface structure should remain largely unchanged and the only decreased N signals should be originated from the in-situ formation of NH²⁻ vacancy presents on the surface of SrNH.

It is not a good idea to compare TOF among literature reports unless active site is well defined. There are a number of ways people calculate TOF. Reporting activity data based on the mass of catalyst is better.

Response: We appreciate the referee point this out. We agree the reviewer's viewpoint that "There are a number of ways people calculate TOF". We believe that the mass activity is the most direct and effective way to demonstrate the activity of a catalyst. Therefore, we first compared the mass activity between our Co/SrNH and other catalysts in Table S1 and S2. While, we also cannot ignore the role of TOF in distinguishing the intrinsic activity of active sites. And then we employed a unified approach to calculate the number of active sites of the loaded metal and determined the TOF of the Co/SrNH as well as other catalysts in Table S3 and S4.